# Gene Polymorphisms *LEP*, *LEPR*, *5HT2A*, *GHRL*, *NPY*, and *FTO*-Obesity Biomarkers in Metabolic Risk Assessment: A Retrospective Pilot Study in Overweight and Obese Population in Romania

Ovidiu Nicolae Penes [1], Bernard Weber [2], Anca Lucia Pop [3,*], Mihaela Bodnarescu-Cobanoglu [4], Valentin Nicolae Varlas [1,*], Aleksandru Serkan Kucukberksun [3], Dragos Cretoiu [1], Roxana Georgiana Varlas [1] and Cornelia Zetu [4]

1  Faculty of Medicine, "Carol Davila" University of Medicine and Pharmacy, 37 Dionisie Lupu Street, 020021 Bucharest, Romania; ovidiu.penes@umfcd.ro (O.N.P.); dragos.cretoiu@umfcd.ro (D.C.); roxana-georgiana.bors@drd.umfcd.ro (R.G.V.)
2  Institute of Medical Virology, University Hospital, 60596 Frankfurt am Main, Germany; bernard.weber@labo.lu
3  Faculty of Pharmacy, "Carol Davila" University of Medd Pharmacy, 6 Traian Vuia Street, 020945 Bucharest, Romania; serkan.kucukberksun@drd.umfcd.ro
4  National Institute for Diabetes and Metabolic Diseases "N. Paulescu", 030167 Bucharest, Romania; mihaelabdn@gmail.com (M.B.-C.); cora.zetu@paulescu.ro (C.Z.)
*  Correspondence: anca.pop@umfcd.ro (A.L.P.); valentin.varlas@umfcd.ro (V.N.V.)

**Abstract:** Genome-wide association studies (GWAS) have successfully revealed numerous susceptibility loci for obesity. The PREDATORR study (2014) shows that in Romania, 346% of adults aged 20–79 y/o are overweight, and 31.4% are obese with a high risk of cardiometabolic complications, a number that puts almost 67% of Romania's population in the abnormal weight group. Our study aims to investigate the current status of the genetic foundation in metabolic disease associated with obesity, applied to a pilot group of patients specifically examining the impact of known polymorphisms and their haplotype of six food intake-regulating genes, namely leptin (*LEP*), leptin receptor (*LEP*-R), serotonin receptor (*5HTR2A*), ghrelin (*GHRL*), neuropeptide Y (*NPY*), and fat-mass and obesity-associated protein (*FTO*) with the following polymorphisms: *LEP A-2548G*, *LEPR A-223G*, *5HTR2A G-1439A*, *GHRL G-72T*, *NPY T-29063C*, *FTO A-T*, and body mass index (BMI). A notable link between the *LEP-2548 rs7799039* gene's AG genotype and the risk of obesity was observed, particularly pronounced in males aged 40–49, with an approximately seven-fold increased likelihood of obesity. The *5HTR2A rs6311 AA* genotype was associated with a higher BMI, which was not statistically significant. The *FTO rs9939609* gene's AA genotype emerged as a significant predictor of obesity risk. Besides these significant findings, no substantial associations were observed with the *LEPR*, *5HTR2A*, *GHRL*, and *NPY* genes. Haplotype association analysis showed a suggestive indication of GRGMLA (*rs7799039*, *rs1137101*, *rs6311*, *rs696217*, *rs16139*, *rs9939609* sequence) haplotype with a susceptibility effect towards obesity predisposition. Linkage disequilibrium (LD) analysis showed statistically significant associations between *LEP* and *LEPR* gene (*p* = 0.04), *LEP* and *GHRL* gene (*p* = 0.0047), and *GHRL* and *FTO* gene (*p* = 0.03). Our study, to the best of our knowledge, is one of the very few on the Romanian population, and aims to be a starting point for further research on the targeted interventional strategies to reduce cardiometabolic risks.

**Keywords:** genes; *LEP*; *LEPR*; *GHRL*; *5HTR2A*; *NPY*; *FTO*; obesity; overweight; biomarkers; genotype; SNP; risk; cardiometabolic

## 1. Introduction

Obesity is a chronic disease, a pandemic with one out of eight individuals worldwide having a body mass index (BMI) $\geq 30 \text{ kg/m}^2$ that has multifactorial determinants related to

lifestyle (sedentary lifestyle, inadequate eating habits) with a ground of genetic, hereditary, psychological, cultural, and ethnic factors [1,2].

Today, a growing body of evidence underlines the influential role of the genetic factors in determining individual susceptibility to weight gain [3]. Genetic profiling has emerged as a promising tool for the therapeutic and nutritional management of metabolic premorbid states, consisting of disturbances of the body's ability to regulate and utilize energy [4]. These mechanisms precede the onset of metabolic disorders such as obesity, type 2 diabetes, and cardiovascular disease.

Lately, genome-wide association studies (GWAS) have identified several gene variants that predispose a person to being overweight and obesity; however, these risk variants explain only a modest proportion of the genetic basis of obesity [5]. Genetic polymorphisms, which represent variations in specific genes, are one missing piece of the obesity puzzle [6]. The analysis of these genetic differences and their impact on BMI has become the new target in obesity research, holding the potential to improve our understanding of obesity's underlying mechanisms and offer more precise, individualized interventions using innovative targeted therapies [7,8] or natural biomolecules [9]. Genetic profiling methods, including whole-genome sequencing, targeted genotyping, and gene expression analysis, effectively identified the genetic variants associated with metabolic premorbid states [10–12].

A genetic variation within a gene associated with obesity is known as a single nucleotide polymorphism (SNP). During the cell's DNA replication process, occasional 'typos' can occur, leading to these variations in the DNA sequence at specific sites [13,14]. SNPs are variations in a single base pair within the DNA sequence, which can arise in genes related to specific conditions, such as obesity. These SNPs have the potential to alter gene function or expression, thereby impacting various metabolic processes, appetite regulation, and energy expenditure. The classification of SNPs is based on their occurrence frequency in the population. 'Wild type' (WT) is the most prevalent form of a gene in a population; this is followed by the 'heterozygous' type (HET). The 'mutant' type (MUT), occurs in less than 0.1% of the population. Each SNP type plays a distinct role in genetic variation and can contribute differently to the study of obesity and related metabolic disorders [15].

The clinical interpretation of these variations requires careful consideration, ideally integrating them with other clinical and lifestyle data for a comprehensive analysis. For a thorough understanding and accurate application of this genetic information, it is advisable to seek genetic counseling and consult with healthcare professionals or genetic specialists who are well-versed in this field.

Leptin and leptin receptor gene polymorphisms may contribute to the individual variability in obesity risk, body weight regulation, and response to weight-loss interventions [16,17]. The leptin receptor gene *LEPR rs1137101* SNP involves a transition from adenine (A) to guanine (G) at codon 223, resulting in an amino acid change from glutamine (Q) to arginine (R). This alteration is thought to impact the signaling capacity of the Lept [18,19].

The 5-HT2A receptor is instrumental in the early differentiation of human primary subcutaneous preadipocytes into adipocytes. Throughout this differentiation process, the 5-HT2A receptor actively influences the expression of the crucial genes that drive adipogenesis [20]. The SNP in the *5-HT2A* gene represents a specific change in the DNA sequence, which is integral to the biological mechanisms previously discussed. In particular, the 1438 GG genotype of this SNP has been linked to variations in the waist-to-hip (W/H) ratio and central adiposity [21]. Moreover, research has indicated a link between the *rs6311 (A1438G)* polymorphism in the *5-HTR2A* gene and the development of eating disorders [22].

The *GHRL* polymorphism rs696217 involves a guanine (G) to thymine (T) substitution, resulting in an amino acid change from leucine (Leu) to methionine (Met) at position 72 in exon 2 of the *GHRL* gene (L72M amino acid change corresponding to the G72T SNP). This alteration in the ghrelin hormone, secreted by enteroendocrine cells in the stomach and binding to the growth hormone secretagogue receptor (GHSR), may influence metabolic

processes and obesity. Mutations in the *GHRL* gene, responsible for regulating ghrelin hormone production—a key player in appetite control—have been shown to heighten the risk of developing metabolic disorders, obesity, and type 2 diabetes [23].

SNPs in the *NPY* gene, linked to neuropeptide Y and its interactions with serotonin pathways, play a significant role in regulating appetite and energy balance [24,25]. These genetic variations in the *NPY* gene are associated with obesity, dietary preferences, and the metabolism of glucose and lipids. Variants of the *NPY* gene, specifically rs16139 and rs17149106, have been linked to a higher BMI and an elevated risk of obesity in individuals of Caucasian European descent, affecting both genders. Additionally, the rs16139 variant, known for its functional impact, is associated with an increased tendency for weight gain starting from a young age [26,27].

The fat mass and obesity-associated protein (*FTO*), alternatively referred to as alpha-ketoglutarate-dependent dioxygenase, is an enzyme encoded by the *FTO* gene located on chromosome 16 [28]. Among its 21 identified SNPs, 18 have been strongly linked to obesity, especially in populations of European descent [29]. The *FTO* SNP rs9939609 was significantly associated with obesity risk. Numerous studies have demonstrated that individuals carrying the risk allele (typically the A allele) of rs9939609 are more likely to have a higher BMI and an increased likelihood of obesity than those with the non-risk allele [30,31]. This association has been consistently observed across various populations but is particularly pronounced in individuals of European descent [32]. The *FTO* gene, including the rs9939609 SNP, is considered one of the most robust genetic predictors of obesity risk identified to date [33]. It was documented that body weight was 3–4 kg higher, and the obesity risk was 1.67 times higher in those patients who were homozygous (AA genotype) for the risk allele than those who were not [34].

Given the notable variability in individual biological responses to increased food intake or decreased energy expenditure, and considering the growing popularity of genetic studies, it has become essential to investigate the genetic influences on body weight [35]. Particularly in our geographical region, there is a lack of published data exploring the connections between genetic nucleotide polymorphisms and their role as predisposing factors in cardiometabolic alterations [36]; this region exhibits a significant prevalence of being overweight and obesity, with rates of 21.1% and 21.8%, respectively [37]. These figures are noteworthy when compared to European statistics, where being overweight and obesity affect nearly 60% of adults and approximately one in three children [38].

This study researched obesity-related genetic polymorphisms, scrutinizing how these genetic variations within our study participants are correlated to BMI and the susceptibility to obesity. We incorporated insights from the latest research, underscoring the potential of these findings in crafting personalized weight loss strategies and combating the global obesity epidemic. The burgeoning field of cardiometabolic and obesity genetics is emerging as a pivotal avenue for advancing their management in the 21st century toward a personalized intervention.

## 2. Materials and Methods

In the present study, we retrospectively analyzed the prevalence of specific genetic polymorphisms and correlation with the body weight profile (body weight and body mass index) for six genes involved in body weight and metabolic balance by processing the genetic test data for 55 patients (female n = 30, male n = 25), in the order of presentation, with age 16–68 years, referred to our clinic for cardiometabolic risk evaluation and weight control (Age Management Clinic®, Bucharest, Romania, EU) (Figure 1).

The exclusion criteria were as follows: patients with cancer, liver and kidney failure, hypothyroidism and hyperthyroidism, Type 2 diabetes mellitus, and those with alcohol and substance addiction were not included in the study. All the data used in the study were previously approved by the patients (or legal tutors) before testing and processing according to the laboratory standards and the current regulations (Ethics Committee approval C4/2022).

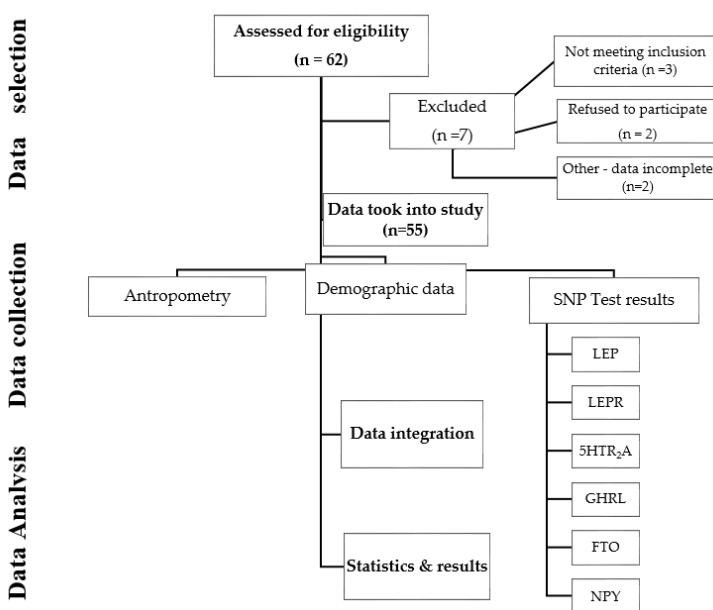

**Figure 1.** Study diagram.

Anthropometric parameters were directly measured by reference to standard protocols. Anthropometric measurements of body weight (kg) and body height (cm) were taken, and BMI (body mass index) was calculated using the "body weight/height$^2$" (kg/m$^2$) formula. Height was measured in cm, with the feet close together and the head in Frankfort plane using a stadiometer with 0.1 cm accuracy. The body weight was measured using the Tanita BC 545 N Inner Scan$^®$ (Balance TM$^®$, Tokyo, Japan) while the subjects were hungry and wearing light clothes and no metal. According to the BMI classification of the World Health Organization (WHO), the BMI values of the participants were grouped into four categories: underweight (BMI < 18.5 kg/m$^2$), normal weight (18.5–24.9 kg/m$^2$), overweight (25.0 kg/m$^2$–29.9 kg/m$^2$), and obese (30.0 ≤ BMI kg/m$^2$).

### 2.1. DNA Extraction and Genotyping

Genotyping was performed using the SNP (Single Nucleotide Polymorphism) scan genotyping protocol, based on a double ligation reaction and multiplex fluorescence PCR, using salivary samples (LRJ$^®$, Luxembourg, EU). We investigated the *LEP*, *LEPR*, *5HTR2A*, *GHRL*, *NPY*, and *FTO* genes (Table 1). We employed in the analysis the following genotypes: wtwt = wildtype (absence of the variation); wtvt = heterozygous genotype (presence of the variation); vtvt = homozygous variant genotype (presence of the variation). Genotype susceptibility can be associated with a negative or positive impact on weight control as described as follows: red = increased risk, yellow = susceptibility associated with the average population, and green = protective genotype or decreased susceptibility.

**Table 1.** Investigated genes and corresponding SNPs and amino acid sequence change.

| Gene | Chromosome | SNP | rsID | Amino Acid | Biochemical Structure |
|------|-----------|-----|------|-----------|----------------------|
| *LEP* | 7 | A-2548G | rs7799039 | - | Leptin |
| *LEPR* | 4 | A-223G | rs1137101 | Q223R | Leptin receptor |
| *5HTR2A* | 13 | G-1439A | rs6311 | - | Serotonin 2A receptor |
| *GHRL* | 3 | G-72T | rs696217 | L72M | Ghrelin |
| *NPY* | 4 | T-29063C | rs16139 | L7P | Neuropeptide Y |
| *FTO* | 16 | A-T | rs9939609 | - | Fat mass and obesity protein |

rs—rs ID number, a unique identifier for a specific SNP in a gene; AA—amino acid change in the specified position; A—adenine, G—guanine; T—timine; amino acids—L-leucine, M—methionine, P-Q—glutamine, R—arginine, L72M—methionine for leucine at the 72nd position—in Ghrelin protein.

*2.2. Statistical Analysis*

We statistically analyzed the data using the Statistical Package for the Social Sciences IBM SPSS Statistics®, version 20 (Armonk, New York, NY, USA). We calculated descriptive statistics, the means and standard deviations for continuous variables or frequencies, and percentages for categorical variables. The association between the presence or absence of an allele and categories of BMI was analyzed using a chi-square ($\chi$2) statistical test. Testing for Hardy–Weinberg equilibrium was performed on each gene. SNP Stats (Inst. Català d'Oncologia, Barcelona, Spain, EU) [39] was used to test the statistical models and analyze haplotype associations.

The clinical characteristics were reported in respect of the frequencies and percentages for the categorical variables and the means and standard deviations for the continuous variables. Categorical variables were analyzed using the chi-square test. The odds ratio (OR) with a 95% confidence interval (95% CI) was reported for allelic comparisons and BMI. We used logistic regression for allele data as the predictor (independent variable) and BMI as the outcome (dependent variable), with BMI dichotomized (obese vs. non-obese). Logistic regression is suitable for modeling the probability of an event (e.g., being obese) occurring given the presence or absence of a specific allele.

## 3. Results

*3.1. Socio-Demographic Data*

The study included 55 participants, 54.5% females (n = 30) and 45.5% males (n = 25). The mean age in both groups was 37.4 +/− 12 y/o. Regarding body mass index, 23.6% (n = 13) of patients had a BMI within the normal weight limits (18.5–24.9 kg/m$^2$), while 41.8% of the total study group (n = 23) were overweight (BMI 25.0 kg/m$^2$–29.9 kg/m$^2$), and 34.6% were obese with a BMI $\geq$ 30.0 kg/m$^2$ (n = 19) (Figure 2).

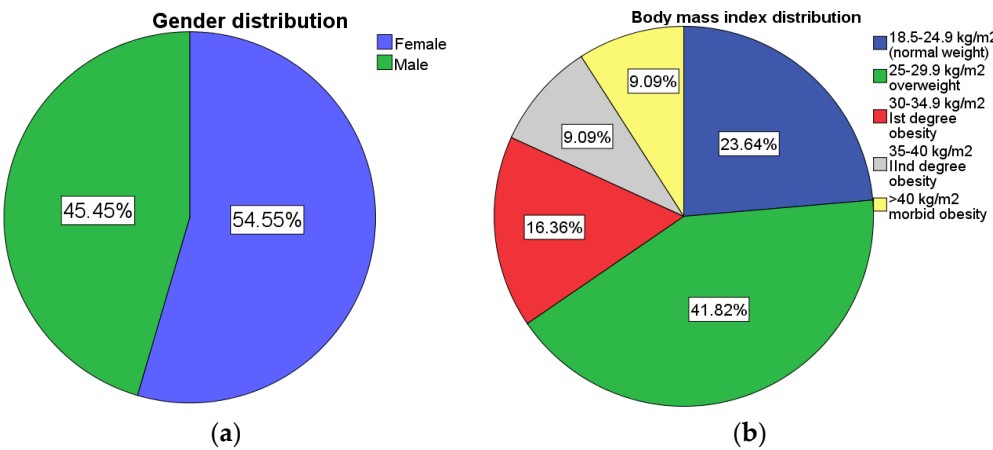

**Figure 2.** (**a**) gender distribution; (**b**) weight distribution of the studied group.

The mean BMI in females was 29.1 kg/m$^2$ +/− 8.37 (X +/− SD), while the mean BMI in males was 29.6 kg/m$^2$ +/− 8.06 (X +/− SD); these results suggest a difference in mean BMI between genders, with females having a slightly lower mean BMI than males (Table 2). All studied gene polymorphisms satisfied the Hardy–Weinberg equilibrium as follows: *LEP* rs7799039 [$p = 0.78$], *LEPR* rs1137101 [$p = 0.78$], *5HTR2A* rs6311 [$p = 0.091$], *GHRL* rs696217 [$p = 1$], *NPY* rs16139 [$p = 1$] and *FTO* rs9939609 [$p = 0.59$].

In the studied group, both genders had a predominant heterozygote (wtvt) variant of *LEP* gene polymorphism: 52.7% (n = 29, 14 females, 15 males) ($p = 0.9$, r = 0.001), with no statistically significant difference between groups (Tables 3 and 4).

**Table 2.** Study group's characteristics.

| Total (N) | | 55 | |
|---|---|---|---|
| Age (X ± SD, y/o.) | | 37.4 ± 12 | |
| Gender (n) | | M = 25, F = 30 | |
| Weight (X ± SD, kg) | | 87 ± 27 kg | |
| Height (X ± SD, cm) | | 172 ± 11.4 cm | |
| BMI (X ± SD, kg/m$^2$) | | 29.4 ± 8 kg/m$^2$ | |
| BMI (kg/m$^2$) | 18.6–24.9 | 25–29.9 | >30 |
| BMI subgroups (N/%) | 13 (23.6%) | 23 (41.8%) | 19 (34.6%) |

BMI—body mass index; y/o—years old; X ± SD—average ± standard deviation.

**Table 3.** Study population gene characteristics.

| Gene | SNP | Genotype Frequencies | | | * Susceptibility | | |
|---|---|---|---|---|---|---|---|
| | | wtwt | wtvt | vtvt | increased | intermediate | decreased |
| *LEP* (n/%) | rs7799039 | 8 (14.5%) | 29 (52.7%) | 17 (30.9%) | 32 (**58.2%**) | 9 (16.4%) | 13 (23.6%) |
| *LEPR* (n/%) | rs1137101 | 17 (30.9) | 29 (52.7%) | 9 (16.4%) | 38 (**69.1%**) | 17 (30.9%) | 0 |
| *5HTR2A* (n/%) | rs6311 | 23 (41.8%) | 20 (36.4%) | 12 (21.8%) | 32 (**58.2%**) | 23 (41.8%) | 0 |
| *GHRL* (n/%) | rs696217 | 45 (81.8%) | 9 (16.4%) | 1 (1.8%) | 0 | 45 (81.8%) | 10 (18.2%) |
| *NPY* (n/%) | rs16139 | 53 (96.4%) | 2 (3.6%) | 0 | 2 (3.6%) | 53 (**96.4%**) | 0 |
| *FTO* (n/%) | rs9939609 | 15 (27.3%) | 25 (45.5%) | 15 (27.3%) | 40 (**72.7%**) | 15 (27.3%) | 0 |

Allele type: wtwt = wildtype (absence of the variation); wtvt = heterozygous (presence of the variation); vtvt = homozygous variant (presence of the variation). rs—rsID number, a unique identifier for a specific SNP in a gene. * Genotype susceptibility can be associated with a negative or positive impact on weight control as described: red = increased risk, yellow = susceptibility associated with the average population, and green = protective genotype or decreased susceptibility.

**Table 4.** Allele predominance and frequency in the studied population.

| Gene | Allele Frequencies | |
|---|---|---|
| *LEP* | A = 46 (43%) | G = 62 (57%) |
| *LEPR* | Q = 63 (57%) | R = 47 (43%) |
| *5HTR2A* | G = 66 (60%) | A = 44 (40%) |
| *GHRL* | L = 100 (91%) | M = 10 (9%) |
| *NPY* | L = 108 (98%) | P = 2 (2%) |
| *FTO* | T = 55 (50%) | A = 55 (50%) |

*3.2. LEP Gene*

*LEP* gene A2548G mutation alleles AA, GA, and GG showed no statistically significant difference in regard to gender. The test for interaction in the trend was not significant ($p = 0.42$), suggesting that the trends observed in the association between genotypes and the response variable did not significantly differ between the genders. The most prevalent *LEP* genotype observed in our study was *LEP* GA (N = 28, 52% of all participants). Men had a five times higher incidence of carrying the GA genotype. OR = 4.9 {95%CI [1.4–16]}. The incidence of male gender, with the GA genotype, was the highest in the 40–49 y/o age group (n = 8, 34.8%).

When we analyzed the correlation between BMI (body mass index) and the incidence and type of *LEP* genotype A254G SNP (AA, AG, or GG), the results were as follows: the GG genotype was associated mainly with normal weight (11.3%) while the GA genotype correlated with overweight and obese patients (47.2%). For the GA genotype in the *LEP* gene, the coefficient was 1.950 with a standard error of 0.845, associating with a statistically significant risk of obesity ($p = 0.02$), with OR = 7.031, CI 95% [1.3–36.8], consistent with the literature data that found a higher prevalence of the GA genotype in overweight and obese patients [40]. However, when we adjusted the data for gender and age, statistical significance was lost. These data provide a closer look at the relationship between the GA genotype susceptibility within the study population and underline the need for an

extensive population study. The association between the presence of the GG allele in the *LEP* gene A2548G SNP and the risk of obesity was not statistically significant ($p = 0.06$), with an OR = 0.269, 95/CI [0.067–1.073].

### 3.3. Leptin Receptor Gene

The A223G SNP had the highest prevalence in the overweight group (n = 14, 28%), but there was no statistically significant difference between weight groups ($p = 0.4$).

In our analysis of the potential association between the *LEPR* AG, AA, and GG genotypes and obesity risk, binary logistic regression indicated that the *LEPR*-A223G genotype's presence is not a statistically significant predictor of obesity risk ($p = 0.4$), with an odds ratio (OR) of 0.571, 95% CI [0.145–2.247]. Neither the AA nor the GG genotype of *LEPR*-A223G showed a statistically significant association with obesity risk ($p = 0.7$ and $p = 0.2$, respectively). However, patients with the AA genotype had a 2.5-fold higher risk of obesity compared to those with the AG or GG genotypes.

### 3.4. 5HTR2A Gene

Statistical analysis of *5HTR2A*-G1439A gene with the GA, AA, and GG genotypes and BMI for five inherited models that analyzed the relationship between each genotype and BMI adjusted for gender and age showed a significant association in three models of five. In codominant model analysis, the AA genotype had a mean BMI of 33.5 kg/m$^2$ exhibiting a significant difference with OR = 5.54 (95% CI: 0.37 to 10.71) compared to GG, but with a *p*-value of 0.06. The GA genotype showed no significant difference regarding BMI compared to GG (GA mean BMI = 27.9 kg/m$^2$).

The log-additive model suggests a significant linear trend ($p = 0.088$) that indicates a trend in BMI increase, with each additional copy of the A allele. The recessive model also showed a significant association of the AA genotype compared with the GG and GA genotypes [mean BMI = 33.5, OR = 5.91 (95% CI: 1.11 to 10.72)].

### 3.5. GHRL Gene

In our study, the GG allele of the *GHRL* gene mutation showed the highest prevalence in overweight patients (N = 19, 38.8%), yet no statistically significant difference was observed between the weight groups. Associations between *GHRL* gene mutation polymorphisms GT ($p = 0.3$), GG ($p = 0.4$), and obesity risk were not statistically significant. The GG polymorphism group had an OR of 1.875, which was not statistically significant.

The genotype analysis of the rs696217 SNP confirmed no association for alleles, genotypes, or any other forms of genetic models.

### 3.6. NPY Gene

Leucine (L) to Proline (P) amino acid change at position 7 (L7P) in the Neuropeptide Y related to the SNP in the *NPY* gene had the highest incidence of the LL allele (N = 21, 42.9%) in the overweight patients' group, with a $p = 0.2$. In the group with the *NPY* LL genotype, an OR = 4.1 was recorded without statistical significance ($p = 0.3$). The genotype analysis of the rs16139 SNP confirmed no association for alleles, genotypes, or any other forms of genetic models.

### 3.7. FTO Gene

In our study examining the *FTO* A-T genotypes in relation to BMI and the risk of being overweight and obesity, the analysis of homozygote T, homozygote A, and heterozygote TA alleles revealed the highest prevalence in the heterozygote group (N = 10, 20.4%) within the overweight patient category ($p = 0.1$). However, no statistically significant association was found between the presence of the *FTO* (rs9939609) TA allele (OR = 2.4) and the risk of obesity ($p = 0.2$). Conversely, the *FTO* (rs9939609) homozygote A allele was a significant predictor of obesity risk (B = 1.48, S.E. = 0.713, Wald = 4.3, df = 1, $p = 0.03$), indicating a fourfold higher risk of obesity compared to the heterozygote or homozygote T alleles (OR=,

95% CI [1.08–17]). The results suggest that, after adjusting for Id, gender, and age, the *FTO* genetic variant was not significantly associated with BMI in this study population. Also, none of the measured models showed statistically significant differences.

The test for interaction in the trend yielded a *p*-value of 0.021, suggesting a statistically significant interaction effect between the *FTO* genetic variant and gender on the BMI response variable after adjusting for age. This implies that the effect of the *FTO* genetic variant on BMI significantly differs between females and males [41].

Linkage disequilibrium (LD) analysis (the statistical method used in genetics to study non-random associations or correlations between alleles at different loci on a chromosome) showed statistically significant associations between the *LEP* and *LEPR* genes (*p* = 0.04), *LEP* and *GHRL* genes (*p* = 0.0047), and *GHRL* and *FTO* genes (*p* = 0.03). The findings provide insights into the patterns of LD between these genetic loci, helping to understand the potential genetic associations and interactions between gene polymorphisms. No strong correlation has been found via LD analysis between alleles at different loci on a chromosome. Moderate or weak correlations have not been shown (Figure 3).

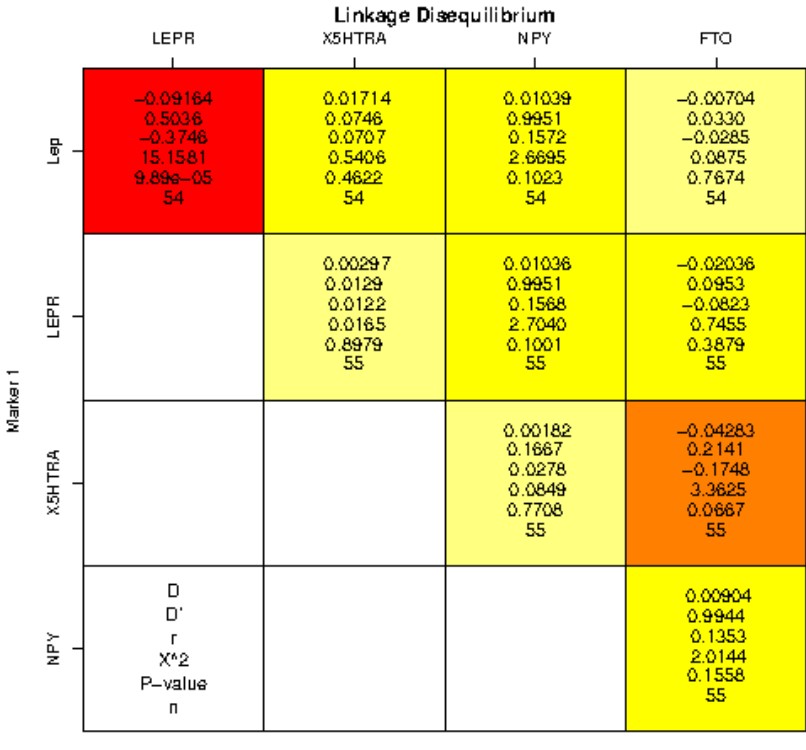

**Figure 3.** Linkage disequilibrium analysis.

The association between haplotypes and obesity was analyzed via logistic regression.

Haplotype association analysis showed a suggestive indication of GRGMLA haplotype (rs7799039, rs1137101, rs6311, rs696217, rs16139, rs9939609 sequence) with a susceptibility effect towards obesity predisposition [*p* < 0.0001, OR = 30.53 (18.3–42.7)] with a 0.0091 (rare) frequency (Table 5). Also, ARGLLT haplotype (rs7799039, rs1137101, rs6311, rs696217, rs16139, rs9939609 sequence) showed the lowest incidence towards obesity predisposition *p* = 0.03, [OR = −9.27 (−17.9–0.58)] with a 0.0205 (moderate) frequency.

**Table 5.** Genotype and codon mutation incidence for each studied gene; statistical significance between the groups of each variable.

| 0 | SNP | Genotype 1 | Genotype 2 | Genotype 3 | *p*-Value |
|---|---|---|---|---|---|
| *LEP* | rs7799039 | AA = 9 (16.4%) | GA = 28 (52%) | GG = 17 (30.9%) | *p* < 0.001 |
| *LEPR* | rs1137101 | QQ = 17 (30.9%) | QR = 29 (52.7%) | RR = 9 (16.4%) | *p* < 0.001 |
| *5HTR2A* | rs6311 | AA = 12 (21.8%) | GA = 20 (36.4%) | GG = 23 (41.8%) | *p* < 0.001 |
| *GHRL* | rs696217 | LL = 45 (81.8%) | LM = 9 (16.4%) | MM = 1 (1.8%) | *p* < 0.001 |
| *NPY* | rs16139 | LL = 53 (96.4%) | LP = 2 (3.6%) | PP = 0 (0%) | *p* < 0.001 |
| *FTO* | rs 9939609 | AA = 15 (27.3%) | TA = 25 (45.5%) | TT = 15 (27.3%) | *p* < 0.001 |

## 4. Discussion

Single nucleotide polymorphisms are recognized as the leading cause of human genetic variability and are a valuable resource for mapping complex genetic traits. Understanding the difference between genomics and genetics, how various single nucleotide polymorphisms (SNPs) add risk or benefit to an individual, and how we can use the results of these tests to improve metabolic health in our patients was the purpose of our research.

The primary purpose of the study was to look at the association of *LEP* (A-2548G), *LEPR* (rs1137101), *5HTR2A* (rs696217), *GHRL*N (rs696217), *NPY*, and *FTO* (rs9939609) SNPs (single nucleotide polymorphisms) as identified in the introduction with BMI since there is today a vast amount of data around obesity-inducing or -associated gene mutation candidates for non-syndromic obesity. We confirmed some findings of a previously studied SNP regarding the leptin gene and its receptor associated with BMI. The *LEP* − 2548 GA genotype is one of the most studied human leptin gene (*LEP*) SNPs in different populations and clinical contexts [42]. Our results correlate with the findings in the research literature that associated common obesity phenotypes with the presence of the *LEP* − 2548 AG genotype [43–45]. Like previous studies (Bilge et al., 2021) [46], our association with the *LEP*-2548 GA genotype was a predictor of obesity, close to being statistically significant in the male gender. These findings suggest that the presence of the GA allele in the *LEP*-A2548-G gene may be a potential risk factor for becoming overweight or obese and the associated complications. Further research and a higher number of patients are necessary to elucidate the exact mechanisms underlying this genetic association and its implications for understanding obesity and preventing cardiometabolic complications.

The results of the *LEP* GG genotype binary logistic regression analysis suggest a weak, not statistically significant association between the presence of the GG allele in the *LEP* gene and a reduced risk of obesity. Patients with the GG genotype appear to be approximately 0.269 times less likely to be obese compared to those without this allele. However, it is important to note that the *p*-value (*p* = 0.06) is above the traditional significance threshold, indicating that this association should be interpreted in the context of our study population. While some studies have identified a significant association of the GG homozygote genotype with obesity in the American population, indicating potential ethnic or geographic variances in the gene's obesity link, the associations between the *LEP*-2548 GG genotype and obesity have not demonstrated a significant link in our population, indicating possible modulation through additional genetic, environmental, or demographic factors, underscoring the variability and complexity of genetic influences on obesity across different demographic groups [47].

*LEPR* (Leptin receptor gene) polymorphisms did not correlate statistically significantly with overweight or obese patient groups and did not predict a higher risk of obesity in our population study. Patients with the AA genotype in the *LEPR* gene had a 2.5-fold higher risk of obesity compared to those with AG or GG genotypes (although not statistically significant).

In the HERITAGE cohort study, the MM genotype for the ghrelin/preproghrelin variants was associated with the lowest BMI (*p* = 0.020). In our study, the effect of genotypes (LL, LM, MM) of the GHRL (rs696217) polymorphism on the BMI of obese patients did not show reliably significant differences; however, the LL genotype had the highest incidence

(81.8%), a fact that can suggest a possible association with obesity in the Romanian population. Also in our study, the MM genotype had the lowest incidence associated with a high BMI (MM = 1.8%). Despite the *5HTR2A* rs6311 AA genotype patients group having almost a six times higher rate of presenting as overweight or with obesity when it was adjusted for age and gender, this result did not present statistical significance with a *p*-value close to significant (*p* = 0.06) [48]. Although the values did not show a statistically significant *p*-value, the mean BMI value in the AA genotype was higher than the AG or GG genotypes, which suggests that the AA genotype is related to a high BMI; we look forward to analyzing a higher number of patients for a better analysis. Other studies reported the association of the A allele with eating disorders and increased BMI, including German and British patients. Also, research data showed no association between *5HTR2A* rs6311 genotypes and obesity [49]. Binary regression analysis suggests that the *FTO* (rs9939609) AA genotype is a significant predictor of obesity risk, with individuals carrying this polymorphism being more likely to be obese.

Haplotype association analysis showed a suggestive indication of GRGMLA haplotype with a susceptibility effect towards obesity predisposition and associated cardiometabolic complications. Although the results showed a rare frequency (0.0091), this haplotype had almost a 30 times higher rate of association with being overweight and obesity.

The use of personalized nutrition to optimize the diet for patients based on genetic variation and knowing the results of the specific diet will be a valuable tool for physicians as well for patients also. Further research, particularly focusing on specific subgroups and considering ethnic and geographic factors, is needed to fully understand the role of the studied gene polymorphism in obesity [50].

Our study has limitations. The study had a limited number of participants or lacked diversity in terms of age, ethnicity, or geographic location; this could limit the generalizability of our findings. The sample size was limited to 55 participants included in a pilot study; however, enhanced data are to be analyzed during a broader investigation. The serum *Leptin* and ghrelin level determinations were not available in this study group. Even if the participants were randomly selected, in order of presentation, there might be selection bias considering their approach towards a health enhancement age management program. Other genes and environmental factors that might contribute to obesity were not considered, which could provide a more comprehensive understanding. The study had a cross-sectional design by establishing some associations without determining the cause; thus, longitudinal studies would be needed to confirm these findings over time.

## 5. Conclusions

Our study underscores the complex relationship between genetic polymorphisms and obesity, suggesting the need for further research to understand the contributing factors better. The study found a significant relationship between the AG genotype in the *LEP*-A2548G SNP and obesity risk. Individuals with this allele were about seven times more likely to be obese. The male gender showed a five-fold higher risk of carrying the high-risk obesity allele, particularly in the 40–49 age group. In contrast, the GG allele in the same gene was not significantly associated with obesity risk.

Regarding the *LEPR*-Q223R SNP, no significant associations with obesity risk were found across different genotypes. Similarly, the *5HTR2A*-G1439A SNP, *GHRL* L72M SNP, *NPY* L7P SNP, and *FTO* rs9939609 SNP gene polymorphisms showed no significant associations with obesity risk, despite some trends observed. The *FTO* rs9939609 SNP AA genotype emerged as a significant negative predictor of obesity risk, indicating a decreased likelihood of obesity in individuals with this allele.

These results highlight the intricate relationship between genetic factors and obesity, revealing a complex interplay and underscoring the need for further research to deepen our understanding of these connections. With new technologies that allow us to understand how genes modulate the body's response to nutrition or how nutrition modulates the

body's response to mutated genes, scientists are solving the mysteries of metabolic health and primary prevention of cardiometabolic complications.

**Author Contributions:** Conceptualization, O.N.P., B.W. and A.L.P.; Data curation, A.L.P., C.Z. and A.S.K.; Formal analysis, O.N.P., B.W., M.B.-C., A.S.K. and R.G.V.; Investigation, O.N.P., B.W. and C.Z. Methodology, O.N.P., A.L.P., M.B.-C. and V.N.V.; Project administration, O.N.P. and A.L.P.; Resources, O.N.P. and B.W.; Software, M.B.-C.; Supervision, D.C. and C.Z. Validation, O.N.P., A.L.P. and C.Z. Visualization, B.W., V.N.V., D.C., R.G.V. and C.Z. Writing—original draft, O.N.P., A.L.P. and M.B.-C.; Writing—review & editing, A.L.P., M.B.-C. and V.N.V. All authors have read and agreed to the published version of the manuscript.

**Funding:** This research received no external funding.

**Institutional Review Board Statement:** Ethical Committee approval C4/10.2022/Retrospective noninterventional study.

**Informed Consent Statement:** Informed consent was obtained from all the subjects involved in the study.

**Data Availability Statement:** The data presented in this study are available on request from the corresponding author.

**Conflicts of Interest:** The authors declare no conflicts of interest.

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
