# Peer review of "Gene Polymorphisms LEP, LEPR, 5HT2A, GHRL, NPY, and FTO-Obesity Biomarkers in Metabolic Risk Assessment: A Retrospective Pilot Study in Overweight and Obese Population in Romania"

_cardiogenetics, doi:10.3390/cardiogenetics14020008_

Round 1

Reviewer 1 Report

Comments and Suggestions for Authors

Please unify the way you present p-values in the abstract (please correct p = 1e-04).

Line 23: it should be status instead of staus

Line 287: There is an unnecessary full stop after “GG” before the bracket.

Line 329: Unnecessary full stop before the bracket.

All of the gene names should be written in cursive throughout the whole manuscript (including the title, abstract, keywords, main text, figures and tables).

Table 3 is the only table with gene names written in a bold print. Please unify the way you present the gene names in tables.

Table 4: Please replace the “N/O” sign with PP = 0 (0%) to unify the presentation of genotypes.

The number of study participants (n=55) is too small. BMI is an imprecise parameter and not the best obesity biomarker, Fat-Free Mass (FFM) or body fat percentage would be better indicators. The materials and methods section seems to be lacking of the information about the sex of the participants (it is, however, mentioned in the results section). What is more, the age range of the participants is between 16 and 68 years, which is a very big difference.

The genotype frequencies of analyzed polymorphisms in the European population are:
GHRL rs696217 GG 84%, GT 16%, TT 1%
NPY rs16139 TT 92,4%, CC 0,4%, CT 7,2%

The number of study participants should be larger to be able to see these genotype distributions. The study group also consists of males and females, who have different distributions and norms for body fat. Please consider analyzing Fat%, FFM and FMI. 

The bibliography should be unified (differences for example between 36, 37 and 39 - please check and unify all of the 49 citations).

Author Response

 The authors thank the reviewer for the thorough evaluation.

  1. Please unify the way you present p-values in the abstract (please correct p = 1e-04).

Answer: Thank you for your observation. We corrected the issue as pointed

  1. Line 23: it should be status instead of status

Answer: Thank you for your observation. We corrected the issue as pointed

  1. Line 287: There is an unnecessary full stop after “GG” before the bracket.

Answer: Thank you for your observation. We corrected the issue.

  1. Line 329: Unnecessary full stop before the bracket.

Answer: Thank you for your observation. We corrected the issue as pointed

  1. All of the gene names should be written in cursive throughout the whole manuscript (including the title, abstract, keywords, main text, figures and tables).

Answer: Thank you for your observation. We corrected the issue as pointed

  1. Table 3 is the only table with gene names written in a bold print. Please unify the way you present the gene names in tables.

Answer: Thank you for your observation. We corrected the issue as pointed

  1. Table 4: Please replace the “N/O” sign with PP = 0 (0%) to unify the presentation of genotypes.

Answer: Thank you for your observation. We corrected the issue as pointed

  1. The number of study participants (n=55) is too small. BMI is an imprecise parameter and not the best obesity biomarker, Fat-Free Mass (FFM) or body fat percentage would be better indicators.

Answer: Thank you for your observation. Totally agree with your precise observation. For our actual pilot study, we have availability only for BMI anthropometric data. In the following study, we analyzed the body composition anthropometric data for more detailed correlations. However, further studies are needed to recognize cut-off values for body adiposity index (BAI), as a marker of body fatness.

  1. The materials and methods section seems to be lacking of the information about the sex of the participants (it is, however, mentioned in the results section).

Answer: Thank you for your observation. We have filled the data (Line 167)

  1. What is more, the age range of the participants is between 16 and 68 years, which is a very big difference.

Answer: Thank you for your observation. Related to the large age span of the study participants, while setting up the inclusion criteria, we have taken into consideration that age does not majorly influence the genetic profile; in terms of obesity criteria, the BMI classification was similar.

  1. The genotype frequencies of analyzed polymorphisms in the European population are:
    GHRL rs696217 GG 84%, GT 16%, TT 1%
    NPY rs16139 TT 92,4%, CC 0,4%, CT 7,2%

Answer: Thank you for your observation. We added the missing general info about genotype frequencies of analyzed polymorphisms in the European populations in “Discussions” section

  1. The number of study participants should be larger to be able to see these genotype distributions.

Answer: Thank you for your observation. Totally agree with your precise observation. We have done a pilot study on a narrow  data field; the statistical power is low, so we consider the results relevant to the observational level. We have mentioned the small study group as a limitation of the current study. A larger sample will be taken into analysis in further research.

  1. The study group also consists of males and females, who have different distributions and norms for body fat. Please consider analyzing Fat%, FFM and FMI. 

Answer: Thank you for your observation. Totally agree with your precise observation. For our pilot study, we have availability only for BMI anthropometric data; in terms of obesity criteria, the BMI classification was similar for men and women. BAI is complementary to BMI and can be recommended for the estimation of body fat.

  1. The bibliography should be unified (differences for example between 36, 37 and 39 - please check and unify all of the 49 citations).

Answer: Thank you for your observation. References will be corrected

Reviewer 2 Report

Comments and Suggestions for Authors

The manuscript titled: Gene Polymorphisms LEP, LEPR, 5HT2A, GHRL, NPY, and FTO - Obesity Biomarkers in Metabolic Risk Assessment: A Retrospective Pilot Study in Overweight and Obese Population in Romania (cardiogenic-2892160) represents an interesting analysis of specific six food intake-regulating genes: and their polymorphisms. Authors found that the LEP-2548 rs7799039 gene's AG genotype was associated with the risk of obesity (particularly in males) with an approximately sevenfold increased likelihood of obesity, while the FTO rs9939609 gene's AA genotype emerged as a significant predictor of obesity risk. The paper is interesting. However, the data regards a small number of patients.

1.     Can you do a reliable occlusion using 55 participants by measuring SNP?

2.     How did you estimate the adequate amount of patients for your study?

3.     Are your conclusions relatable using such a small amount of analyzed subjects?

Author Response

Review 2

The authors thank the reviewer for the thorough evaluation.

The manuscript titled: Gene Polymorphisms LEP, LEPR, 5HT2A, GHRL, NPY, and FTO - Obesity Biomarkers in Metabolic Risk Assessment: A Retrospective Pilot Study in Overweight and Obese Population in Romania (cardiogenic-2892160) represents an interesting analysis of specific six food intake-regulating genes: and their polymorphisms. Authors found that the LEP-2548 rs7799039 gene's AG genotype was associated with the risk of obesity (particularly in males) with an approximately sevenfold increased likelihood of obesity, while the FTO rs9939609 gene's AA genotype emerged as a significant predictor of obesity risk. The paper is interesting. However, the data regards a small number of patients.

  1. Can you do a reliable occlusion using 55 participants by measuring SNP?

Answer: Thank you for your observation. We totally agree with your observation. We have done a pilot study on a narrow data field; the statistical power is low, but since there was only one case under the age of 18, it does not affect the statistical significance of the batch. We have mentioned the small study group as a limitation of the current study. Regarding the size of the batch, we can mention that since it was a pilot study we could not give up any patient, the statistical relevance is only the partial fact that was stated in the article. A larger sample will be taken into analysis in further research.

  1. How did you estimate the adequate amount of patients for your study? Are your conclusions relatable using such a small amount of analyzed subjects?

Answer: Thank you for your observation. As mentioned previously, our statistical power is not sufficiently relevant for the entire population; however, we found correlations that were statistically significant. This is a pilot study that we are planning to extend regarding the number of participants and clinical and paraclinical measurements. The results have shown us that reliable and important data can be obtained by increasing our number of participants. We consider it the first step to expand our research, as there are no other studies to our knowledge made on Romanian participants.

As we know genetic variability can have unique blueprints in every region of the world, so this is why we consider it important. Pilot studies regarding gene polymorphisms and their relationships with BMI have been made before and their results and the fast development of nutrigenetic inspired us to start small with the data that we had and continue growing our research as number of patients and genes analyzed.

Reviewer 3 Report

Comments and Suggestions for Authors

In the present study authors examined the relationship between SNPs within the selected genes involved in body weight and metabolism regulation, body weight status in the sample of the Romanian population. The topic and the results are of some interest, however, there are also important concerns associated with this study.

1.     The Introduction is too long. The basic data about SNPs of interest could be presented in the table containing data such as the role of encoded protein, implications of polymorphic variants for its expression/activity and relationship with obesity according to previous studies.

2.     The study was performed in small group of patients. Was the required sample size calculated before the study?

3.     The important limitation is that only BMI was calculated. The study would benefit from addressing the distribution of adipose tissue, e.g. abdominal adiposity status such as waist circumference.

4.     The method of genotyping should be described.

5.     BMI should be reported with the precision of 0.1, not 0.01 kg/m2

6.     Was the difference between BMI in males and females significant? If not, it should not be stated that it was different.

7.     The phrase “GA allele” should be corrected to: “GA genotype”.

Comments on the Quality of English Language

English language is quite fine. Only minor language polishing is required.

Author Response

Dear esteemed REVIEWER,

In the present study authors examined the relationship between SNPs within the selected genes involved in body weight and metabolism regulation, body weight status in the sample of the Romanian population. The topic and the results are of some interest, however, there are also important concerns associated with this study.

  1. The Introduction is too long. The basic data about SNPs of interest could be presented in the table containing data such as the role of encoded protein, implications of polymorphic variants for its expression/activity and relationship with obesity according to previous studies.

Answer: Thank you for your observation. Totally agree with your observation. As the complexity of data is new and wide, we have decided to have an explicit introduction. We adjusted the issue as pointed.

  1. The study was performed in small group of patients. Was the required sample size calculated before the study?

Answer: Thank you for your observation. We have done a pilot study on a narrow data field; the statistical power is low; We have mentioned the small study group as a limitation of the current study. A larger sample will be taken into analysis in further research. We consider it the first step to expand our research, as, to our best knowledge, there are no other studies made on Romanian participants on the topic.

As we know genetic variability can have unique blueprints in every region of the world, so this is why we consider it important. Pilot studies regarding gene polymorphisms and their relationships with BMI have been made before and their results and the fast development of nutrigenetics inspired us to start small with the data that we had and continue growing our research as number of patients and genes analyzed.

  1. The important limitation is that only BMI was calculated. The study would benefit from addressing the distribution of adipose tissue, e.g. abdominal adiposity status such as waist circumference.

Answer: Thank you for your observation. Totally agree with your precise observation. For our pilot study, we have availability only for BMI anthropometric data; in terms of obesity criteria, the BMI classification was similar for men and women. BAI is complementary to BMI and can be recommended for the estimation of body fat.

The method of genotyping should be described.

Answer: Thank you for your observation. We described the method at line 192-194

  1. BMI should be reported with the precision of 0.1, not 0.01 kg/m2

Answer: Thank you for your observation. We corrected the issue as pointed

  1. Was the difference between BMI in males and females significant? If not, it should not be stated that it was different.

Answer: Thank you for your observation. Statement added “The median BMI value was of 28,1+/- 8,37 (X+/-SD) in women and 27,5 +/- 8,06 (X+/-SD) in men group. (please see Line 203)

  1. The phrase “GA allele” should be corrected to: “GA genotype”.

Answer: Thank you for your observation. We corrected the issue as pointed

Round 2

Reviewer 1 Report

Comments and Suggestions for Authors

Line 25: You corrected all prefixes „lep” to italics. Please make sure to correct only the names of the genes “LEP” and write “leptin” regularly.

Line 38: p = 1e-04 does not equal 0.04. Make sure you are putting the correct number.

Line 76: I would suggest rather phrasing this sentence as follows: ‘Wild type’ (WT) is the most prevalent form of a gene in a population.’

Line 97: I think you missed the first bracket at the end of the sentence.

Line 118: There is an unnecessary dot before citation [23].

Line 138: You mistakenly deleted the first few letters of the next sentence.

Line 178: You deleted Figure 1 description but the figure remains in the text. If the deleted description needs change, please add a new, improved description.

Table 1 & section 2.3: Please correct LEPtin and LEPtin receptor to Leptin and Leptin receptor

Line 339: Please add the number of table mentioned in the brackets.

Table 5: Please provide a table with better resolution. The text and numbers are not clear.

The bibliography is still not unified. Please compare, for example, number 4 and 5. Number 40 is also written in bigger font than other positions. Numbers 47 and 48 are written in a smaller font.

Author Response

Dear esteemed reviewer,

Thank you for your work,

Please find our corrections made as adressed:

Line 25: You corrected all prefixes „lep” to italics. Please make sure to correct only the names of the genes “LEP” and write “leptin” regularly.

Answer: Thank you for your mention, we have adressed the issue accordingly.

Line 38: p = 1e-04 does not equal 0.04. Make sure you are putting the correct number.

Answer: Thank you for your mention, we have adressed the issue accordingly.

Line 76: I would suggest rather phrasing this sentence as follows: ‘Wild type’ (WT) is the most prevalent form of a gene in a population.’

Answer: Thank you for your mention, we have adressed the issue accordingly.

Line 97: I think you missed the first bracket at the end of the sentence.

Answer: Thank you for your mention, we have adressed the issue accordingly.

Line 118: There is an unnecessary dot before citation [23].

Answer: Thank you for your mention, pint deleted.

Line 138: You mistakenly deleted the first few letters of the next sentence.

Line 178: You deleted Figure 1 description but the figure remains in the text. If the deleted description needs change, please add a new, improved description.

Answer: Thank you for your mention, we have added the mention of the figure in text (Line 146)

Table 1 & section 2.3: Please correct LEPtin and LEPtin receptor to Leptin and Leptin receptor

Answer: Thank you for your mention, we have addressed the issue accordingly.

Line 339: Please add the number of table mentioned in the brackets.

Answer: Thank you for your mention, we have addressed the issue accordingly.

Table 5: Please provide a table with better resolution. The text and numbers are not clear.

Answer: Thank you for your mention, we have addressed the issue. We have replaced the table with the best available resolution the application is permitting, we hope it is better fitted now.

The bibliography is still not unified. Please compare, for example, number 4 and 5. Number 40 is also written in bigger font than other positions. Numbers 47 and 48 are written in a smaller font.

Answer: Thank you for your mention, we have addressed the issue accordingly.

Reviewer 2 Report

Comments and Suggestions for Authors

The manuscript has been sufficiently improve.

Author Response

Dear Esteemed reviewer,

The authors thank you for your work and support

Kindest regards,

from the behalf of all authors,

dr. Anca Pop

Reviewer 3 Report

Comments and Suggestions for Authors

The manuscript has been revised according to the reviewers' comments.

Author Response

(The authors gave the same response as above.)
